# Integration of bioinformatics and identification of the role of m6A genes in NAFLD

**Jianguo Ma**[1☉]**, Rongyi Xu**[1☉]**, Renlin Li**[2]**, Yangyang Fu**[1]**, Jing Xu**[1]**, Lei Zhou**[1]*, **Yan Qi** [1]*

**1** The First Affiliated Hospital of Yunnan University of Traditional Chinese Medicine, Kunming, China, **2** College of Acupuncture and Massage, Yunnan University of Traditional Chinese Medicine, Kunming, Yunnan, China

☉ These authors contributed equally to this work and share first authorship.
* qiyankm@163.com (YQ), 18915511159@163.com (LZ)

## Abstract

Non-alcoholic fatty liver disease (NAFLD) is prevalent worldwide and seriously affects health. M6A methylation is crucial in its pathogenesis. In this study, a thorough analysis of three gene expression datasets identified nine key differentially expressed genes DEGs associated with m6A methylation in NAFLD that are involved in important biological processes. Subsequently, functional enrichment analysis, weighted gene co-expression network analysis (WGCNA), gene set variation analysis (GSVA) and immune infiltration analysis were conducted to explore the molecular mechanism and gene expression patterns. The LASSO risk model contains a total of 5 m6A-related differentially expressed genes (m6A-RDEGs)(RBM15, IGF2BP2, EIF3B, YTHDC1, WTAP), and the diagnostic model based on these key genes has high accuracy. Among them, YTHDC1 and WTAP are used as prominent biomarkers. In addition, an interaction network between mRNA and miRNA, RNA-binding protein (RBP), transcription factor (TF) and drugs is also constructed. Finally, the animal model of NAFLD was successfully established and validated by RT-qPCR and western blot. This study provides a valuable tool for clinical diagnosis and drives the progress of NAFLD research.

## Introduction

Nonalcoholic fatty liver disease (NAFLD) is a frequently observed disorder. It is a typical feature of hepatic fat accumulation in people who consume little or no alcohol. It represents a liver presentation of metabolic syndrome, which is closely related to obesity, dyslipidemia, and type 2 diabetes [1]. Its prevalence is between 20% and 30% globally, making it a frequent risk factor for chronic liver disease. About 25% of the U.S. population is affected by NAFLD, with a significant proportion progressing to nonalcoholic steatohepatitis (NASH), cirrhosis, fibrosis, or even hepatocellular carcinoma [2,3]. Although the prevalence of NAFLD is high, the mechanism underlying its pathogenesis remains unclear, and specific pharmacological therapeutic techniques approved for NAFLD or NASH are lacking. Lifestyle modifications, such as diet and exercise, are the main ways to manage NAFLD, but these strategies are often insufficient, highlighting the need to develop novel therapeutic approaches.

**Data availability statement:** The publicly available datasets (GSE36807, GSE38713,

GSE47908 and GSE231993) used for this study can be found in the GEO database

**Funding:** This research was supported by the National Natural Science Foundation of China (Grant number [82460794]) , Xingdian Talent Support Program-Youth Talent Special Project (2023. No. 166). Data collection and analysis:High-level TCM talents: (reserve talents) project of Yunnan (2021. No. 1) The main role played by the funders was financial support and did not have additional roles in the study design, data collection and analysis, the decision to publish, or the preparation of the manuscript

**Competing interests:** The authors have declared that no competing interests exist.

**Abbreviations:** NAFLD, Nonalcoholic fatty liver disease; GO, Gene Ontology; TF, transcription factor; PCA, principal component analysis; WGCNA, weighted gene coexpression network analysis; GSVA: Gene Set Variation Analysis; MF, molecular function; LASSO, least absolute shrinkage and selection operator.

The effect of N6-methyladenosine (m6A) RNA methylation, a dynamically reversible process catalyzed by writers, erasers, and readers, on various conditions has recently attracted considerable attention. It has the highest prevalence among endogenous eukaryotic messenger RNA (mRNA) modifications [4] and plays a key role in regulating mRNA stability, splicing, translation, and attenuation [5]. Recently, m6A methylation was reported to affect liver metabolism, fibrosis, and inflammation, indicating its effect on the pathogenesis of NAFLD [6]. For example, METTL3 and FTO regulate adipogenesis by influencing m6A methylation and RUNX1T1 levels. Additionally, overexpression of FTO can increase lipid accumulation and decrease m6A methylation levels in adipocytes, whereas the overexpression of METTL3 can inhibit lipid accumulation and increase m6A methylation levels [7–10]. YTHDC2 can bind to adipogenic gene mRNAs and reduce mRNA stability while inhibiting the expression of corresponding genes, thus inhibiting adipogenesis to regulate TG homeostasis and inhibit hepatic steatosis [5]. These findings indicate that m6A methylation is crucial for treating NAFLD.

As m6A RNA methylation is important for NAFLD, in this study, differential m6A-related gene expression in NAFLD patients was assessed, and their biological functions and underlying mechanisms of action were investigated. Data from three independent datasets (GSE89632, GSE37031, and GSE135251) were used, and differential expression gene analysis, weighted gene coexpression network analysis (WGCNA), functional enrichment analysis, gene set variation analysis (GSVA), and immune infiltration analysis were performed. Additionally, mRNA-miRNA, mRNA-RBP, mRNA-TF, and mRNA-drug interaction networks were established to identify possible therapeutic targets. Key findings were validated through animal experiments, including RT-qPCR and Western blotting assays.

Our comprehensive analysis revealed that m6A-related genes, which are related to various biological processes and pathways associated with disease progression, are differentially expressed in NAFLD. These results provided new insights into the molecular mechanisms underlying NAFLD. Targeting m6A RNA methylation can offer a novel therapeutic approach for this prevalent and challenging liver disease.

## Materials and methods

### Data acquisition

We obtained the GSE89632, GSE37031, and GSE135251 datasets, which are relevant to NAFLD, from the Gene Expression Omnibus (GEO) database [11] using the R package GEOquery [12]. The samples in these datasets were collected from human livers (*Homo sapiens*). We specifically profiled GSE89632 on the GPL14951 platform, GSE37031 on the GPL14877 platform, and GSE135251 on the GPL18573 platform. These platforms are summarized in S1 Table.

All datasets comprised samples from individuals with NAFLD and healthy controls. GSE89632 included 39 NAFLD samples and 24 control samples, GSE37031 included eight NAFLD samples and seven control samples, and GSE135251 included 206 NAFLD samples and 10 control samples. This study included all of these samples.

The m6A-related gene set was curated from the PubMed literature, which yielded 97 genes. After merging and removing duplicates, 34 unique m6A-related genes were identified (S2 Table).

To mitigate batch effects, the R package sva was used for processing the datasets GSE89632 and GSE37031, resulting in an integrated GEO dataset including 47 NAFLD samples and 31 control samples. Additionally, principal component analysis (PCA), a dimensionality reduction approach that projects high-dimensional data onto lower-dimensional space, was used to verify batch effect removal efficiency, facilitating visualization in 2D/3D plots.

Finally, the R package limma was used to normalize and annotate the probes of the integrated GEO dataset and GSE135251, ensuring standardized expression data for subsequent analysis.

## Identification of NAFLD-related differentially expressed genes (m6A-RDEGs)

To elucidate the biological pathways related to DEGs and their associated mechanisms in NAFLD, the limma package was used to analyze the differentially expressed genes (DEGs) of the NAFLD dataset and the GSE135251 dataset. DEGs were screened according to the threshold of log fold change (logFC) >0 and adjusted p-value (adj. P. Val) <0.05. Genes with a positive logFC value were considered to be upregulated, whereas those with a negative logFC value were considered to be downregulated.

To identify m6A-RDEGs related to NAFLD, the variance of the NAFLD dataset and GSE135251 were standardized. We then cross-referenced the DEGs with the m6A-related gene set and used a Venn diagram to illustrate these gene set intersections. The R package ggplot2 was used to draw volcano plots, whereas pheatmap was used to generate heatmaps to visualize the results of the DEG analysis.

## Functional enrichment analysis (GO) of differentially expressed genes

We performed Gene Ontology (GO) annotation, which includes biological process (BP), molecular function (MF), and cellular component (CC) terms, on m6A-RDEGs using the R package clusterProfiler. The p-value threshold and the Benjamini-Hochberg (BH) approach were used to correct the p-value to control the false discovery rate.

## WGCNA of the NAFLD dataset

We conducted WGCNA with the R package [13], with RsquaredCut set to 0.85, a minimum of 100 module genes, and a module combined splicing height of 0.4. This approach facilitated the identification of coexpression modules associated with genes from different groups in the NAFLD dataset (Control/NAFLD). The m6A-RDEGs identified through differential analysis subsequently overlapped with module genes exhibiting the highest correlation between the NAFLD group and the control group to identify candidate genes.

## Key gene screening and LASSO risk model construction

The support vector machine (SVM) model was constructed based on the m6A-RDEG levels of the NAFLD dataset to identify the most accurate and least erroneous genes related to m6A (m6A-RDEGs). Subsequently, the least absolute shrinkage and selection operator (LASSO) regression was performed using the R package glmnet, and reproducibility was ensured by setting the number of seeds to 500. LASSO regression, which includes a penalty term to mitigate overfitting, was illustrated through diagnostic and variable trajectory plots. The m6A-RDEGs selected by the final LASSO model were considered to be key genes for subsequent analyses. We computed the risk score using the LASSO risk score formula and stratified the NAFLD dataset into high-risk or low-risk categories based on the median risk score. This methodology was also extended to the GSE135251 dataset. Moreover, the LASSO risk score (RiskScore) was determined as follows:

$$\text{RiskScore} = \sum_i \text{Coefficient}\left(\text{gene}_i\right) * \text{mRNA Expression}\left(\text{gene}_i\right) \tag{1}$$

## Diagnostic performance of key genes and validation of the LASSO risk model

Using the R package rms, a nomogram was constructed based on the LASSO regression results to illustrate the relationships of key gene levels in the NAFLD dataset with the diagnosis of NAFLD. The R package ggDCA [14] was used to produce decision curve analysis (DCA) plots based on these key genes; this approach is used to evaluate molecular biomarkers, perform diagnostic tests, and construct clinical prediction models. Using similar methods, another nomogram was constructed to depict the relationship between key gene expression in the GSE135251 dataset and NAFLD diagnosis. Subsequently, ggDCA was used to produce DCA plots based on these key genes.

## Immune infiltration analysis

To assess the proportions of various immune cell infiltrates, we used a single-sample gene set enrichment analysis (ssGSEA) algorithm. The enrichment scores derived from the GSVA package in R reflected the presence of various immune cells among different samples [15,16].

Boxplots were drawn to illustrate differences in the infiltration levels of immune cells between the NAFLD and control groups. We determined Pearson correlation coefficients between two groups within this dataset and graphically represented them using the R package ggplot2.

The NAFLD dataset's gene expression matrix was integrated to calculate and present correlations between immune cells and key genes through correlation point plots with ggplot2. Additionally, ssGSEA was applied to all genes in the NAFLD dataset to analyze enrichment differences and parameters consistent with those used for the NAFLD and control groups were used.

## m6A phenotype score

The R package GSVA was used to calculate the m6A phenotype score (m6A score) via the ssGSEA algorithm on key gene expression matrices from the NAFLD and GSE135251 datasets, which were associated with NAFLD. All samples were categorized into high-m6A or low-m6A score groups based on the median score; this grouping facilitated the comparison of the m6A score distributions across the datasets.

We also generated receiver operating characteristic (ROC) curves for the m6A score in both datasets using the R package pROC, delineating the performance in the high and low groups. The efficacy of the key genes in reflecting the m6A phenotype score was also assessed using these curves. Then, the area under the curve (AUC) was used to evaluate diagnostic performance; the AUC value ranged from 0.5 to 1. An AUC closer to 1 signifies greater diagnostic efficacy, while AUCs of 0.5–0.7 suggest poor performance, those of 0.7–0.9 suggest moderate performance, and values exceeding 0.9 denote high performance.

## Construction of mRNA-miRNA, mRNA-RBP, mRNA-TF, and mRNA-drug interaction networks

We used the ENCORI database to predict miRNAs potentially interacting with key genes and subsequently constructed an mRNA-miRNA interaction network by selecting pairs that were documented across at least four sources. RNA-binding proteins (RBPs) were identified through the same database, applying criteria of clusterNum ≥ 7 and clipExpNum ≥ 7 to filter mRNA-RBP interaction pairs for network visualization.

Transcription factors (TFs) binding to key genes were determined by CHIPBase (version 3.0) and the hTFtarget database. The Cytoscape software was used to visualize the mRNA-TF interaction network, which helped us understand human health-related information.

This study assessed the CTD database to predict small molecules or drugs that interact with key genes, with the mRNA-drug interaction network visualized using the Cytoscape software for determining potential therapeutic relationships.

## Experimental animals and groups

Our study protocol was approved by the Medical Ethics Committee of the First Affiliated Hospital of Yunnan University of Traditional Chinese Medicine(DW-2024–029). The experiments strictly adhered to animal ethics guidelines. Wild-type male C57BL mice (18–20 g, six weeks old) were raised in a temperature-controlled room (23 ±1 °C) under a 12-h/12-h light-dark cycle and allowed free access to water and food. After acclimating to a standard chow diet for one week, they were randomly assigned to either the normal diet control (CON) group or the high-fat diet (HFD) group (n = 8 mice/group). The animals were maintained on these diets for 12 weeks after NAFLD induction. After this experiment ended, they were euthanized with 1% sodium pentobarbital (3 ml/kg) to collect liver tissue samples for either hematoxylin and eosin (H&E) and Oil Red O staining or immediately flash frozen in liquid nitrogen and later preserved at -80 °C before use.

## Histopathological observation

After immersing in 4% paraformaldehyde for 24 h, the liver tissues were dehydrated, embedded in paraffin, and cut into 5–7-μm slices before H&E staining. To conduct Oil Red O staining, liver samples were frozen in OCT compound, sliced into 8-μm sections, and stained with Oil Red O. Each group included six animals, and microscopic observations were conducted using a Thermo microscope (Waltham, MA, United States) at 100× magnification; four fields of view per sample were examined to assess changes in liver tissue.

## RT-qPCR analysis

Initially, TRIzol (Invitrogen) was used to isolate liver RNA, which was subsequently reverse-transcribed into cDNA with PrimeScript™ RT Master Mix (TAKARA). RT-qPCR analysis was performed using an Agilent MX3000P system with SYBR Premix Ex Taq II (TAKARA). Each reaction mixture consisted of SYBR Premix Ex Taq II (10 μL), the corresponding primers (0.5 μL), cDNA (1 μL), and ddH$_2$O (8 μL). The PCR amplification conditions were as follows: initial denaturation at 95 °C for 30 s; 5 s at 95 °C; and 10 s at 62 °C for 40 cycles. The $2^{-\Delta\Delta Ct}$ method was used to quantify gene expression, with GAPDH used as the reference. The detailed primer sequences are listed in S3 Table.

## Western blotting

Liver tissues were lysed using RIPA buffer (Beyotime, Shanghai, China) containing protease inhibitors for protein extraction. The protein content was analyzed using the BCA approach, followed by 5 min of boiling at 95 °C. Next, 60 μg of protein was loaded per well, followed by 1 h of separation through 12% SDS-PAGE at 110 V. After transferring the proteins to the PVDF membranes, the membranes were immersed in 5% nonfat milk at ambient temperature (20–25 °C) for 2 h. GAPDH(proteintech. 60004-1-Ig), EIF3B(proteintech. 10319-1-AP), IGF2BP2(proteintech. 11601-1-AP), YTHDC1(proteintech.14392-1-AP), RBM15(proteintech. 22249-1-AP), and WTAP(proteintech) primary antibodies were applied to the membranes and incubated overnight at 4 °C. The membranes were washed three times and incubated with

HRP-conjugated secondary antibody(Proteintech. RGAR001) for 1 h, exposed in a standard enhanced chemiluminescence reaction accord ing to manufacturer's instructions (Servicebio. G2020), and the ImageJ software was used to quantify the immunoreactive bands.

## Statistical analysis

The data were processed and analyzed using the R software (version 4.1.2). Continuous data were presented as the mean ± SD and analyzed by the Wilcoxon rank sum test and the Kruskal-Wallis test between two and multiple groups. Categorical data were analyzed by conducting the Chi-square test or Fisher's exact test. The molecular correlation coefficients were computed using Spearman's correlation analysis, with $P < 0.05$ indicating statistical significance.

## Result

### Analysis of metabolism-related differential genes in NAFLD

NAFLD-related genes were comprehensively analyzed using the GSE89632 and GSE37031 datasets. The sva package in R was used to mitigate batch effects, followed by normalization with the limma package to create a unified NAFLD dataset (Fig 1A–1B), encompassing 47 NAFLD and 31 healthy control samples. The PCA results confirmed a substantial reduction in batch effects and improved sample uniformity post-correction (Fig 1C-D). The GSE135251 dataset, comprising 216 samples (206 NAFLD and 10 normal samples), was also standardized to achieve homogenous expression profiles (Fig 1E–1F).

By conducting differential expression analysis, we identified 14,612 DEGs in the NAFLD dataset, with 5,250 genes satisfying the thresholds (|logFC| > 0, adj.P. Val < 0.05). These DEGs included 2,974 upregulated and 2,276 downregulated genes. Similarly, the GSE135251 dataset revealed 15,826 DEGs, of which 7,133 genes met the criteria, with 3,471 upregulated and 3,662 downregulated genes. These results were visualized using volcano plots (Fig 2A-B).

To identify m6A-RDEGs, we initially identified DEGs common to both datasets by making a Venn diagram (Fig 2C). Further intersection with m6ARGs revealed nine m6A-RDEGs, which are illustrated in a Venn diagram (Fig 2D). Their expression variability was assessed by conducting the Wilcoxon signed rank test, which revealed significant differences and substantial significance in the NAFLD and GSE135251 datasets (Fig 2E and 2F). The differential expression profiles of these genes were then elucidated and presented in a heatmap drawn using the R package pheatmap (Fig 2G–2H).

### GO annotation of m6A-RDEGs

To further elucidate the biological implications of nine m6A-rDEGs in NAFLD, GO functional annotation was conducted on these genes (S4 Table). The results of the analysis showed that the above genes were related mostly to the regulation of alternative mRNA splicing, a process intrinsic to the spliceosome, as well as other mRNA metabolic processes, including RNA methylation. Regarding cellular components, the genes were strongly associated with the eukaryotic translation initiation complex and nuclear speck. Regarding molecular functions, the genes were involved in translation initiation factor activity, nucleic acid binding, and RNA binding, among others. The GO results were presented using bar charts (Fig 3A) and a network diagram (Fig 3B-D).

Additionally, an integrated logFC GO enrichment analysis was performed for the nine m6A-rDEGs, and the respective z scores of diverse molecules were calculated based on their logFC values upon differential analysis of the NAFLD dataset. The results of the combined logFC GO enrichment analysis are shown using a bubble plot (Fig 3E); the findings indicate that most of the enriched GO terms were related to BP.

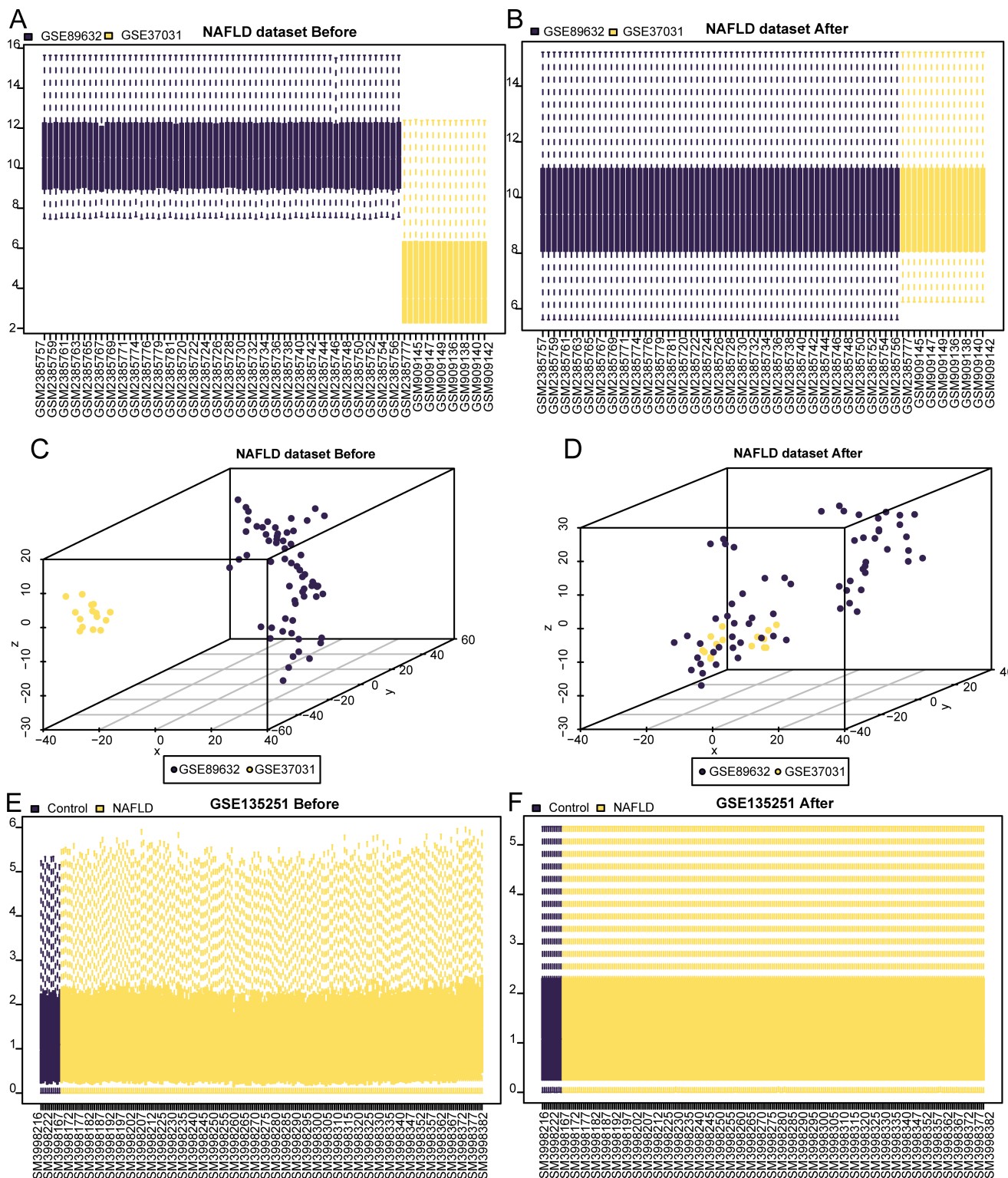

**Fig 1. The NAFLD Dataset and GSE135251 Dataset Were Processed in Batches.** A-b. boxplot of NAFLD dataset before (A) and after (B) batch effect removal treatment. C-d. PCA plot of NAFLD dataset before (C) and after (D) batch effect removal treatment. E-f. boxplot plot of dataset GSE135251 before (E) and after (F) removal of batch effect treatment. PCA: Principal Component Analysis, NAFLD: Nonalcoholic fatty liver disease.

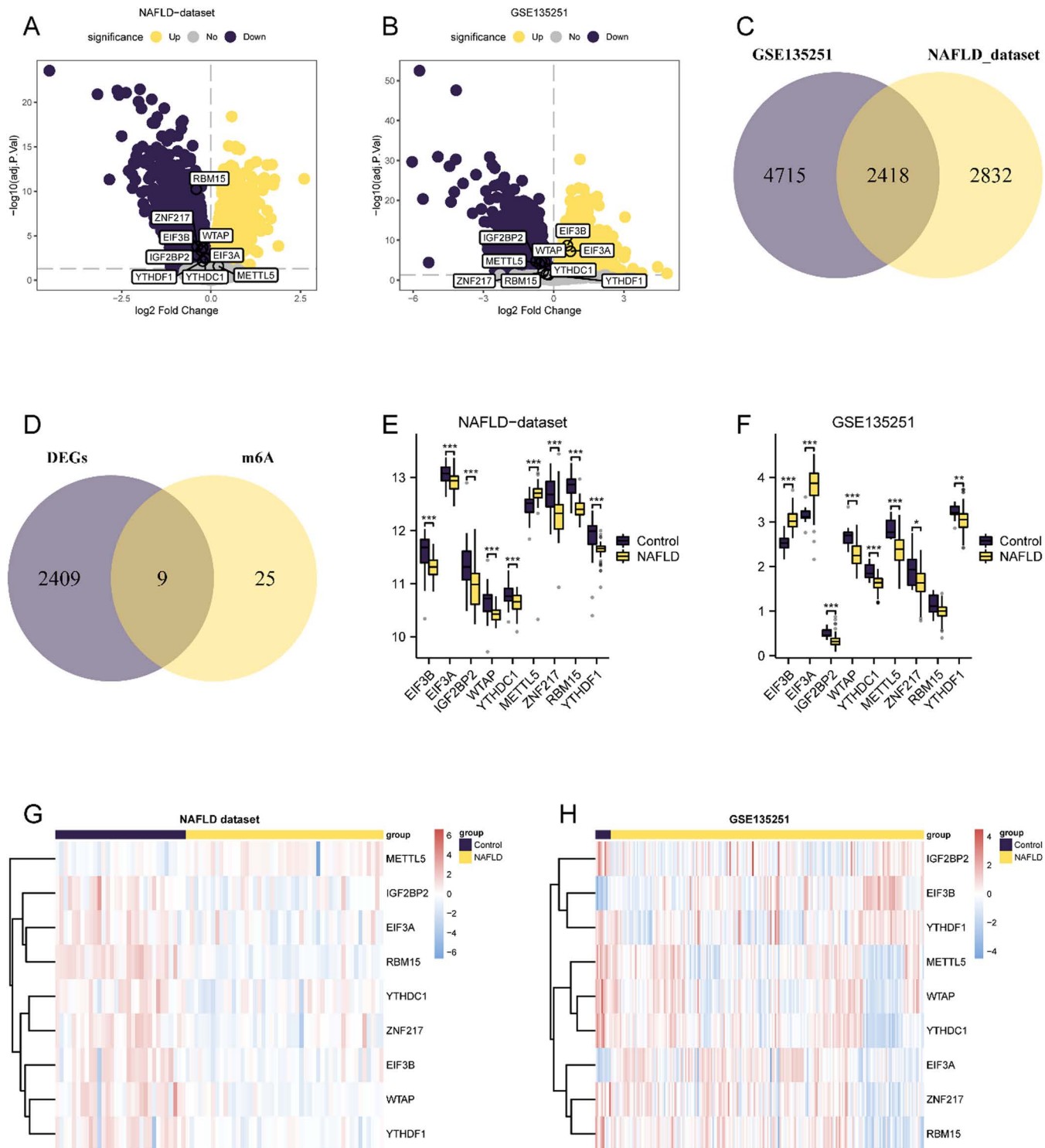

**Fig 2. Differential Analysis of NAFLD Dataset and Dataset GSE135251.** A. Volcano plot of differentially expressed genes analysis of NAFLD dataset. B. Volcano plot of differentially expressed genes analysis of non-alcoholic fatty liver disease group (NAFLD) relative to normal Control group (Control) in dataset GSE135251. C. Venn diagram of differentially expressed genes in NAFLD dataset and differentially expressed genes in dataset GSE135251. D. Venn diagram of common differentially expressed genes (DEGs) and m6A related genes (m6A). E. Differential expression analysis of m6A-RDEGs in NAFLD dataset. F. Differential expression analysis of m6A-RDEGs in dataset GSE135251. G. Simple numerical heatmap of m6A-RDEGs in the NAFLD dataset. H. Simple numerical heat map of m6A-RDEGs in dataset GSE135251.

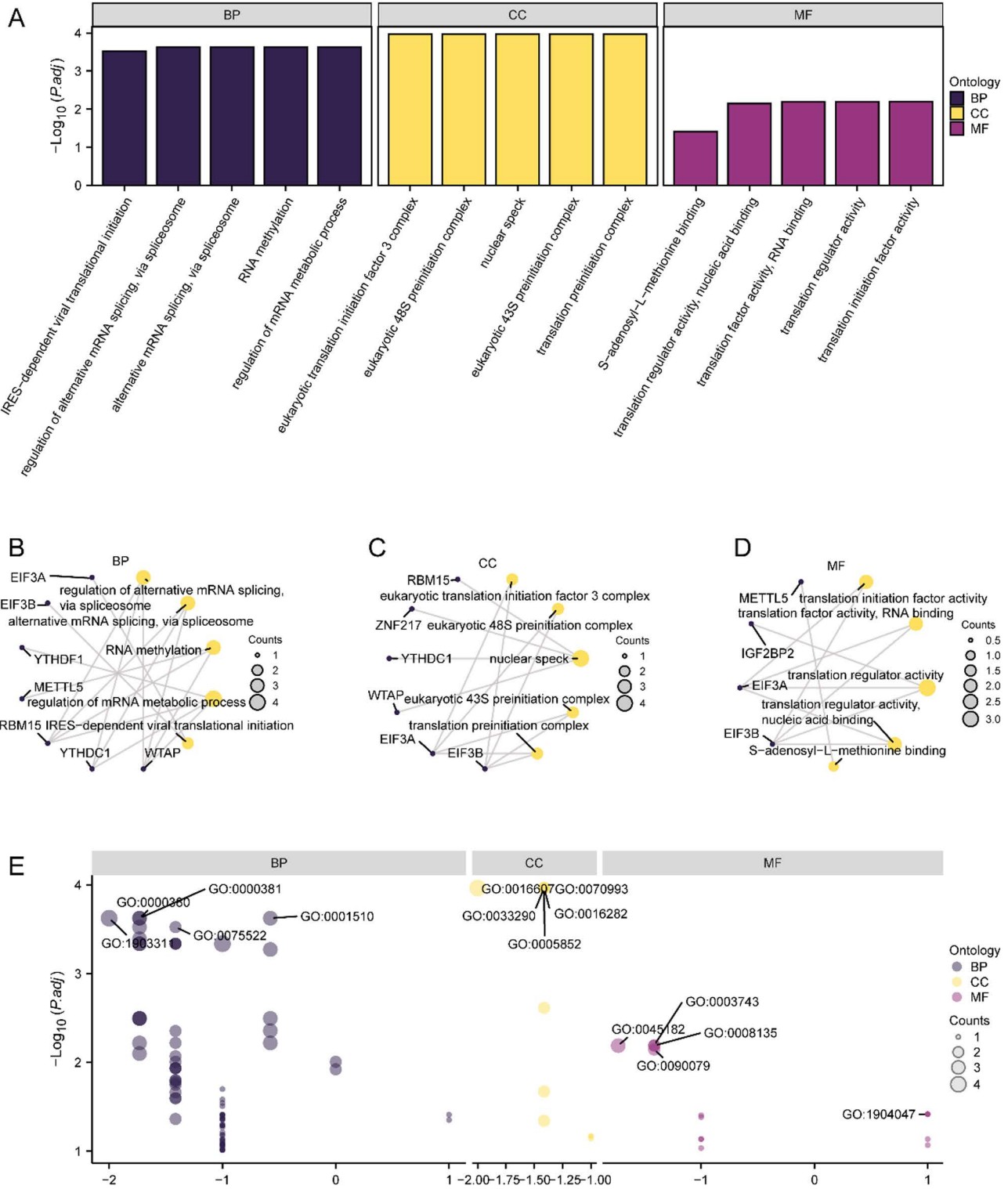

**Fig 3. Functional Enrichment Analysis (GO) of M6A-Related Differentially Expressed Genes.** A. GO functional enrichment analysis of m6A-related differentially expressed genes BP, CC, MF bar chart display. B-d. GO functional enrichment analysis results of m6A related differentially expressed genes BP(B), CC(C), MF(D) network diagram display. E. Bubble plot display of GO enrichment analysis results of m6A-related differentially expressed genes combined with logFC. In bubble plot (E), blue circles represent BP pathways, yellow circles represent CC pathways, and purple circles represent MF pathways. In the network diagram (B-D), yellow dots represent specific pathways and blue dots represent specific genes. The screening criteria for GO enrichment items were p. Adj < 0.05 and FDR value (q.value) < 0.05.

### Differential gene analysis and WGCNA of the NAFLD dataset

To analyze the DEGs of the different groups in the NAFLD dataset (NAFLD vs. control), we conducted WGCNA. The 20% genes with the most significant differences according to mean expression were used to perform hierarchical clustering, which delineated the NAFLD samples, as shown in Fig 4A. Using a power threshold criterion of 0.85, an optimal value of 4 was used for network construction, categorizing genes into MEgray, MEyellow, MEturquoise, MEblue, and MEbrown modules (Fig 4C). A module merge cutoff height of 0.4 was set, facilitating the consolidation of modules with shear heights below this threshold (Fig 4D). The gene expression patterns were correlated with the phenotypic classification of NAFLD subtypes, identifying four key modules (MEyellow, MEblue, MEturquoise, and MEgray) in the dataset (Fig 4E). By intersecting the nine m6A-RDEGs with the MEblue module and subsequently with the MEturquoise and MEyellow modules, six m6A-related genes (METTL5, EIF3B, EIF3A, ZNF217, RBM15, and YTHDF1) were identified, as shown in Fig 4F–4H.

### Key gene screening and LASSO risk model construction

The SVM model, which was developed with nine m6A-RDEGs, identified six genes that maximized the diagnostic accuracy for NAFLD (Fig 5A–5B). These genes were further refined through LASSO regression, yielding a model involving five genes, including *RBM15*, *IGF2BP2*, *EIF3B*, *YTHDC1*, and *WTAP* (Fig 5C–5D). A RiskScore was calculated for the NAFLD and GSE135251 datasets, stratifying NAFLD patients into high-risk or low-risk groups according to the median RiskSore. RiskSore was calculated using the following equation:

$$\text{RiskScore} = (-4.992) * \text{RBM15} + (-0.732) * \text{IGF2BP2} + (-1.031) * \text{EIF3B} \\ + (\text{-}1.375) * \text{YTHDC1} + (-0.658) * \text{WTAP}$$

(2)

We constructed a logistic regression diagnostic model with a nomogram illustrating the effect of key gene expression on the diagnosis of NAFLD, highlighting RBM15, WTAP, and IGF2BP2 as significant contributors (Fig 5F). The results of the DCA confirmed the clinical utility of the model, revealing a superior net benefit over standard benchmarks (Fig 5G). For the GSE135251 dataset, a similar nomogram was constructed, and DCA was performed, based on which YTHDC1, WTAP, and RBM15 were identified as particularly influential, with the model demonstrating higher diagnostic potential (Fig 5H–5I).

### Differential analysis of the ssGSEA immune characteristics of the NAFLD dataset

To further investigate the immune characteristics of key genes in the NAFLD dataset, we used the ssGSEA algorithm to calculate the infiltration levels of 28 immune cell types, such as activated B cells, CD8+ T cells, and CD4+ T cells (Fig 6A), in the NAFLD group and the normal group. The results of the correlation analysis showed a significant positive correlation between the degree of cell infiltration in the NAFLD cohort (Fig 6B). Moreover, five key genes were related to these immune cell infiltration levels, revealing substantial positive relationships at the P < 0.05 threshold (Fig 6C).

### Construction of m6A phenotype scores and correlation analysis of key genes

An m6A phenotype score was derived from the expression of five key genes (RBM15, IGF2BP2, EIF3B, YTHDC1, and WTAP) in the NAFLD dataset and the GSE135251 cohort.

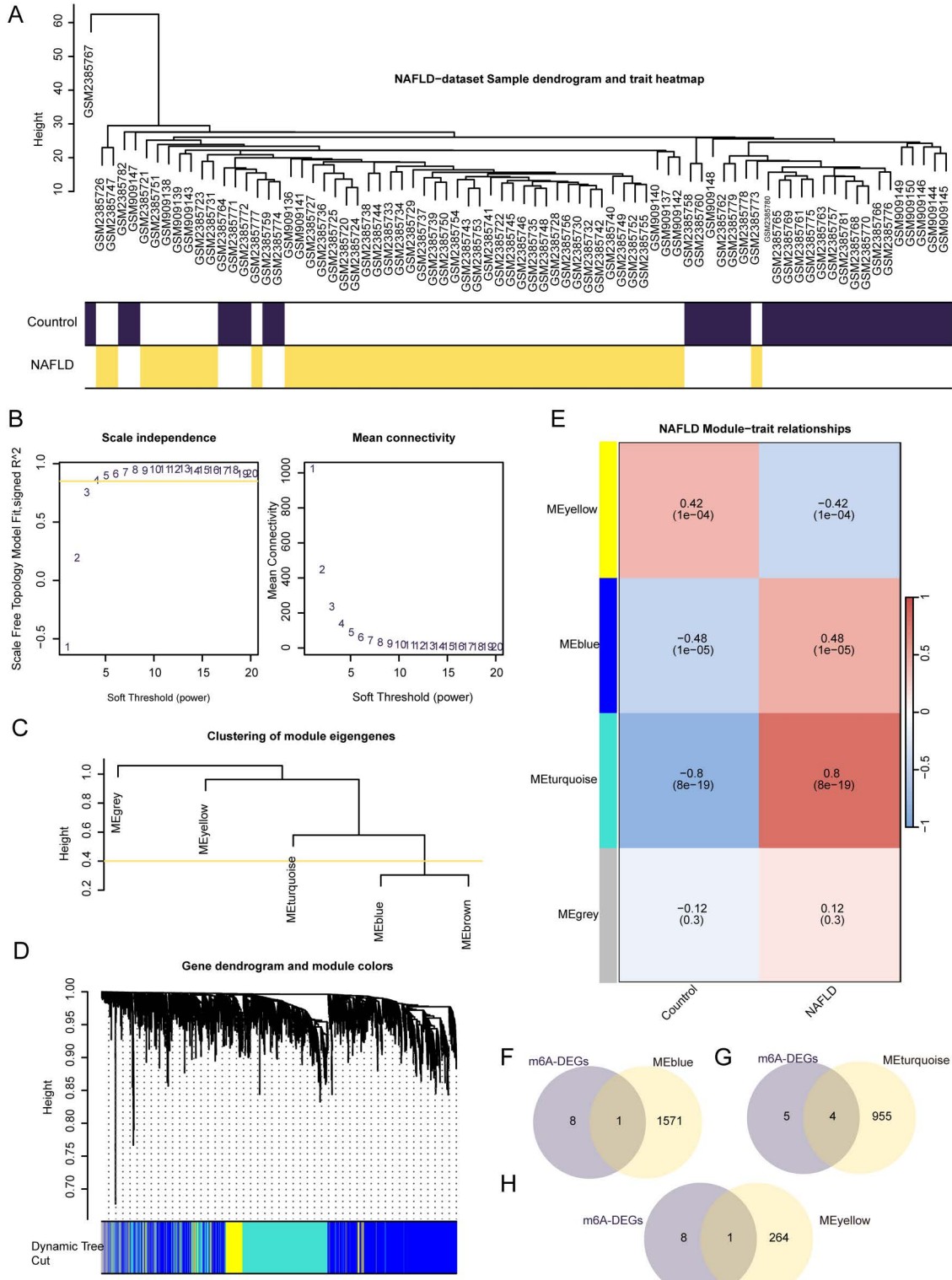

**Fig 4. WGCNA Analysis Identified Co-Expression Modules in NAFLD Dataset.** A. Demonstration of WGCNA analysis clustering of NAFLD dataset samples in dataset. B. Scale-free network display of sample module screening threshold in NAFLD dataset. C. Display of gene module clustering results in NAFLD dataset. D. The corresponding relationship between genes and modules in NAFLD dataset is displayed. E. The results of correlation analysis between NAFLD dataset gene cluster module and different groups are displayed. F-h. Venn diagram of NAFLD dataset M6A-related differentially expressed genes (m6A-RDEGs)

with MEblue (F), MEturquoise (G) and MEyellow (H) module genes. WGCNA: Weighted gene co-expression network analysis. M6a-rdegs: m6A related differentially expressed genes. NAFLD: Nonalcoholic fatty liver disease.

The m6A phenotype score was calculated for each sample using the ssGSEA algorithm. The results of the Mann-Whitney U tests revealed significant differences in the expression of key genes, such as IGF2BP2, YTHDC1, and EIF3B, between groups stratified by the m6A phenotype score in the NAFLD dataset (Fig 7A). Similar findings were recorded for the RBM15, YTHDC1, and WTAP genes in the GSE135251 dataset (Fig 7B).

Diagnostic efficacy was further assessed through ROC curve analysis, which yielded AUC values that suggested substantial diagnostic potential for IGF2BP2, YTHDC1, and WTAP in the NAFLD dataset (Fig 7D–7F). Conversely, RBM15 and WTAP showed lower diagnostic utility (Fig 7C–7G). In the GSE135251 dataset, YTHDC1 and WTAP were found to be strong diagnostic indicators (Fig 7K–7L), in contrast to RBM15, IGF2BP2, and EIF3B, which showed lower diagnostic efficacy (Fig 7H–7J).

## Construction of the interaction network of mRNAs-miRNAs, mRNAs-RBPs, mRNAs-TFs, and mRNAs-drugs

The mRNA-miRNA interactions that involved five key genes were predicted using the starBase database, whereas the Cytoscape software was used to visualize the resulting network (Fig 8A). The network included four key genes (IGF2BP2, EIF3B, YTHDC1, and WTAP) and 43 miRNA molecules, with 44 mRNA-miRNA pairs identified. The detailed interactions can be found in S5 Table. We predicted mRNA-RBP interactions based on the ENCORI database and visualized them (Fig 8B). This network included five key genes (RBM15, IGF2BP2, EIF3B, YTHDC1, and WTAP) and 23 RBP molecules, which formed 58 mRNA-RBP interactions. The specific interactions are detailed in S6 Table. Transcription factor (TF) binding to key genes was identified through the CHIPBase and hTFtarget databases, with interaction data visualized (Fig 8C). The mRNA-TF network included 52 TFs that interacted with two key genes (RBM15 and IGF2BP2), with RBM15 exhibiting 53 mRNA-TF pairs. These interactions are listed in S7 Table. The CTD database was used to identify possible molecular compounds and drugs that target these five key genes, followed by a description of the mRNA-drug interaction network (Fig 8D), which included 61 compounds related to key genes, with specific interactions provided in S8 Table.

## Confirmation of the hub genes in the mice used as a model for NAFLD

Compared to the control group, the model (MOD) group presented significantly higher mouse body weights and an elevated liver index. The total cholesterol (TC) and triglyceride (TG) contents in the liver and serum also increased considerably, and alanine aminotransferase (ALT) and aspartate aminotransferase (AST) activities also increased substantially. Histological analysis of NAFLD mice revealed severe steatosis of hepatocytes, characterized by small round cytoplasmic vacuoles and lymphocyte infiltration around the central vein or within the lobules (Fig 9).

## Validation by RT-qPCR

The results of the RT-qPCR analysis confirmed that the expression of the EIF3B mRNA was elevated in the NAFLD group compared to its expression in the control group, whereas the expression of the RBM15, YTHDC1, and WTAP levels were reduced. These findings were consistent with the results of our bioinformatics analysis (Fig 10).

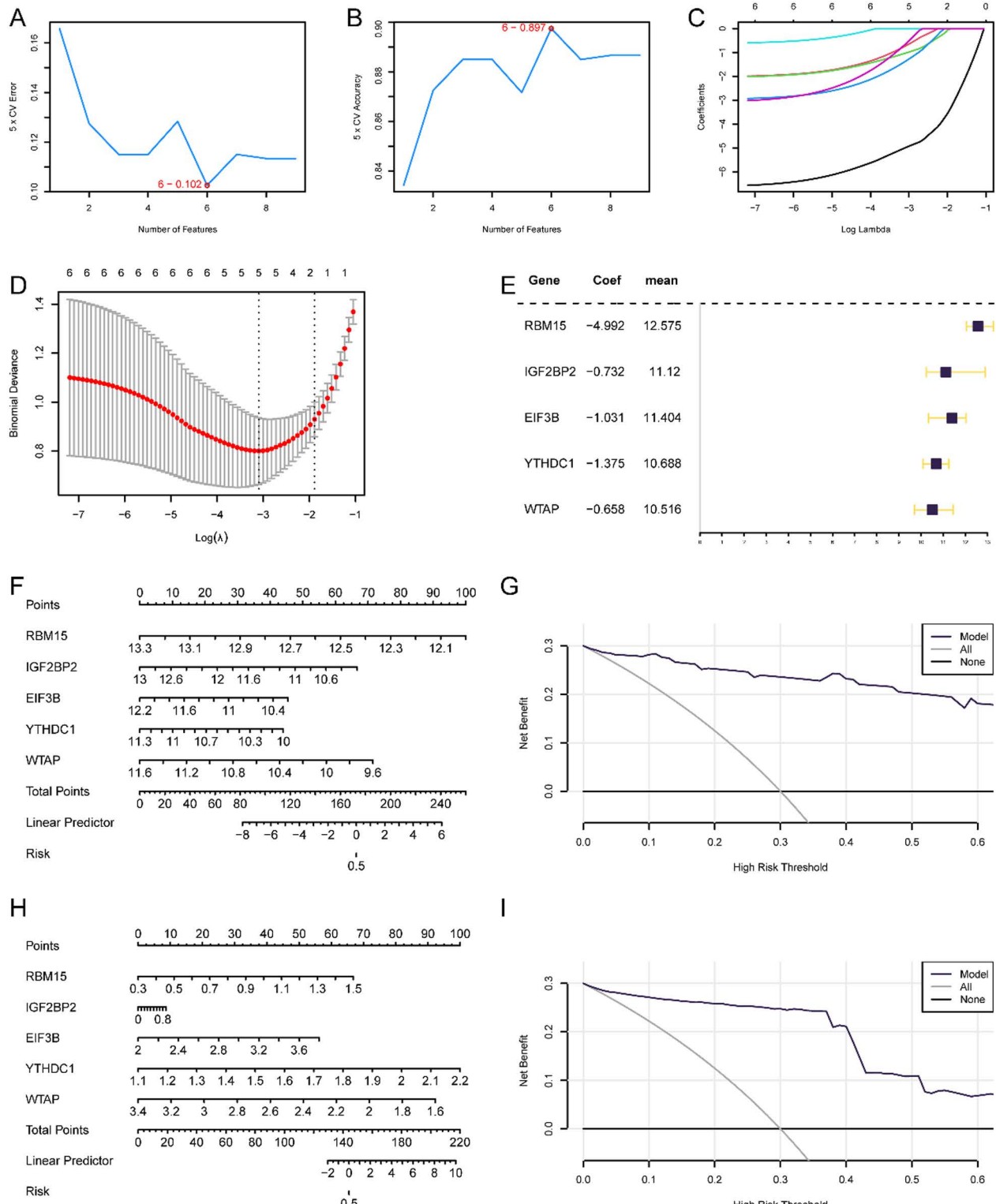

**Fig 5. Key Genes Screening and LASSO Risk Model Construction.** A. The number of genes with the lowest error rate obtained by SVM algorithm. B. The number of genes with the highest accuracy obtained by the SVM algorithm. C. Diagnostic model diagram of LASSO regression model. D. Variable trajectory plot of LASSO regression model. E. Forest plot of key genes in LASSO regression model. F. Nomogram of key genes in NAFLD diagnostic model based on NAFLD dataset. G. Decision curve analysis (DCA) plot of key genes of nonalcoholic fatty liver disease (NAFLD) diagnostic model based on NAFLD dataset. H. Nomogram of key genes in non-alcoholic fatty liver disease (NAFLD) diagnosis model

based on dataset GSE135251. I. Decision curve analysis (DCA) plot of key genes of non-alcoholic fatty liver disease (NAFLD) diagnostic model based on dataset GSE135251. SVM: Support Vector Machine. M6a-rdegs: m6A related differentially expressed genes, NAFLD: Nonalcoholic fatty liver disease, LASSO, Least Absolute Shrinkage and Selection Operator.

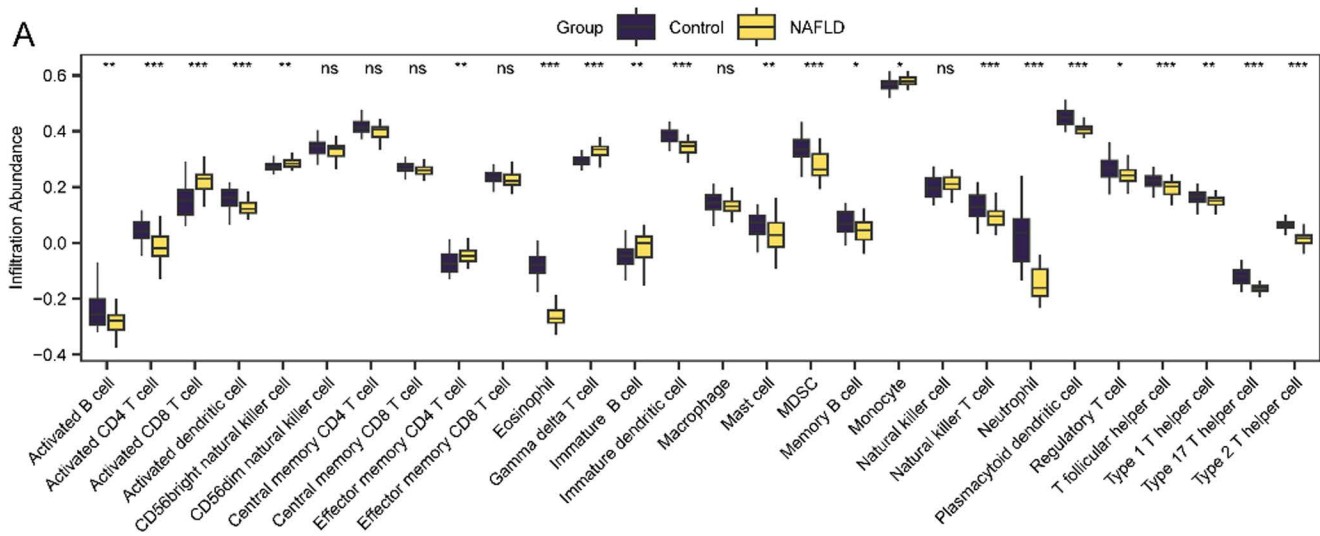

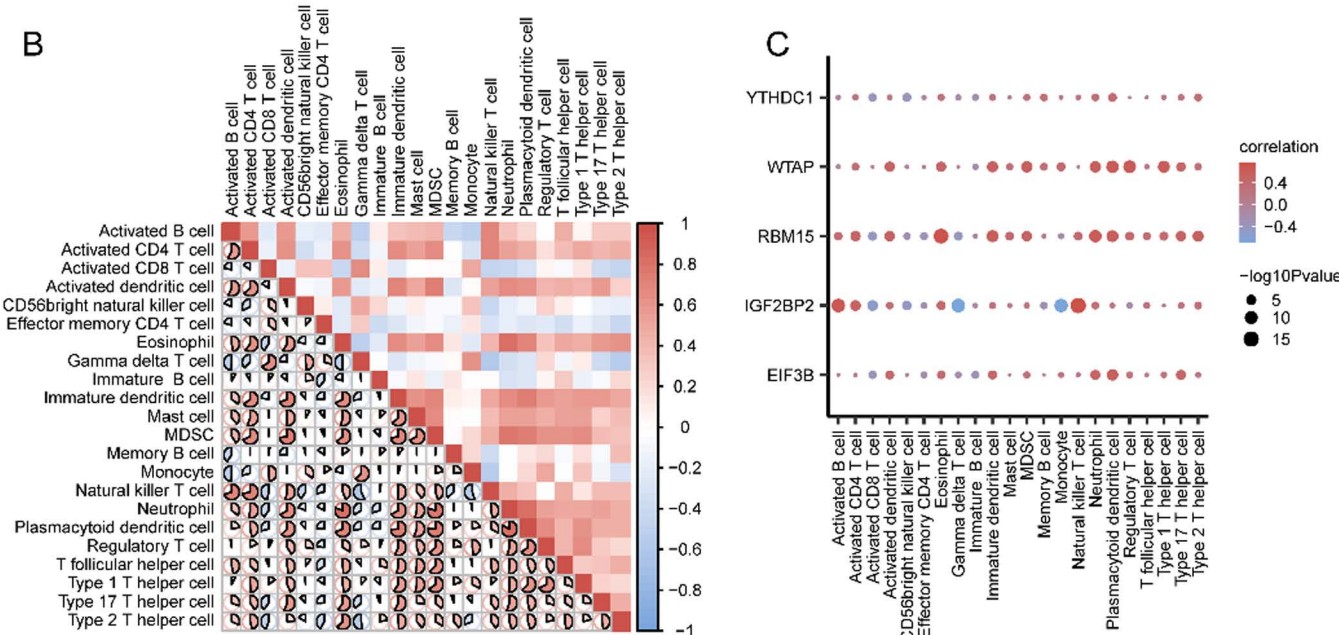

**Fig 6. Differential Analysis of ssGSEA Immune Characteristics of NAFLD Dataset.** A. The results of ssGSEA immune infiltration analysis between the NAFLD group and the normal group in the NAFLD dataset were shown. B. The results of correlation analysis of immune cell infiltration abundance in NAFLD dataset are presented. C. Dot plot of correlation between immune cells and key genes in NAFLD dataset. The symbol ns is equivalent to **P** ≥ 0.05, which is not statistically significant; The symbol * is equivalent to **P** < 0.05 and statistically significant. ssGSEA, single-sample gene-set enrichment Analysis. NAFLD: Nonalcoholic fatty liver disease.

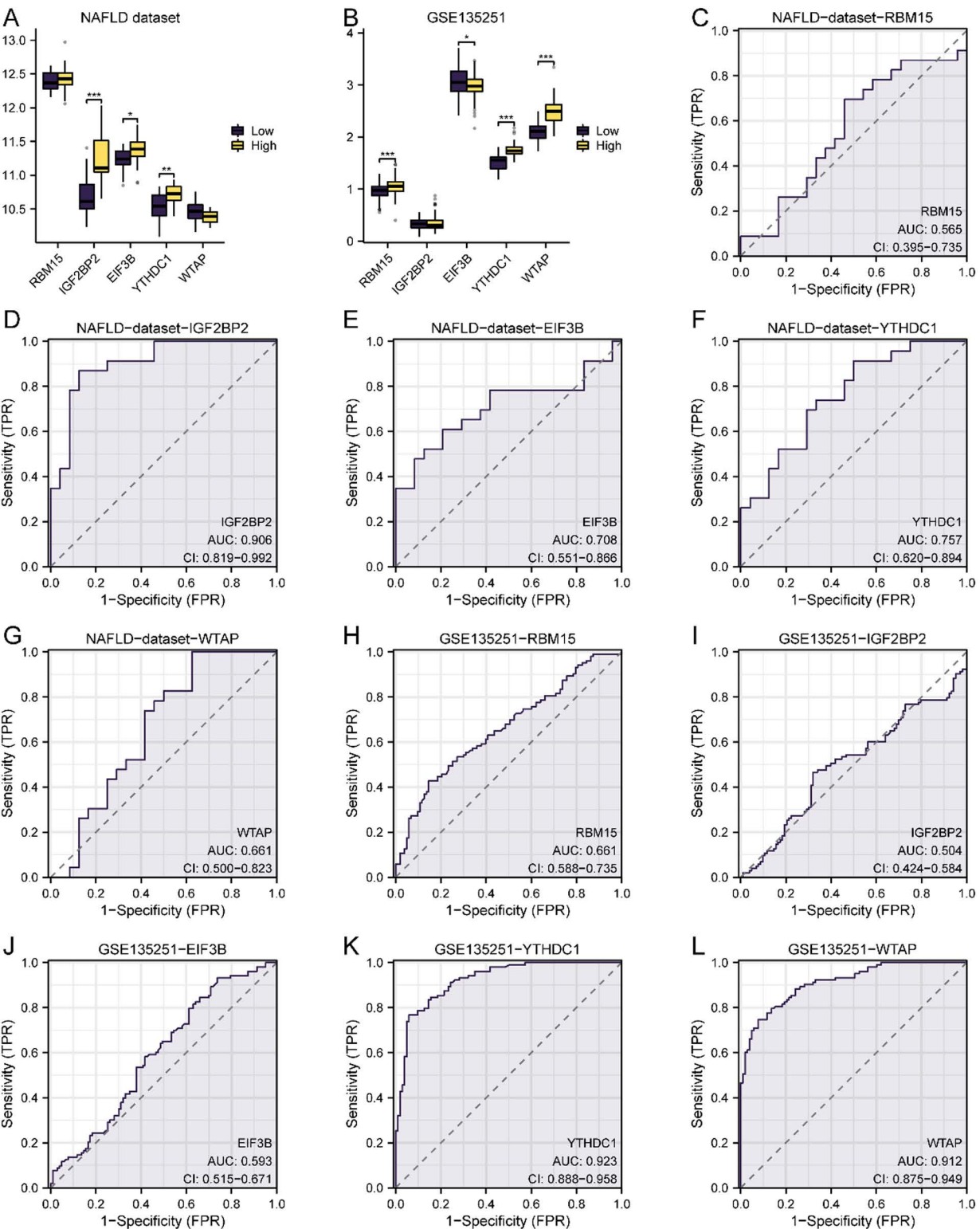

**Fig 7. Construction of m6A Phenotype Score and Correlation Analysis of Key Genes.** A. Group comparison figure of key genes between NAFLD dataset m6A phenotype score Low group and High group. B. Group comparison diagram of key genes between Low group and High group of m6A phenotype score in GSE135251 dataset. C-g. ROC curve results of key genes RBM15 (C), IGF2BP2 (D), EIF3B (E), YTHDC1 (F), WTAP (G) in NAFLD dataset samples with m6A phenotype score Low group and High group are shown. H-l. The ROC curve results of key genes RBM15 (H), IGF2BP2 (I), EIF3B (J), YTHDC1 (K), WTAP (L) in the non-alcoholic fatty liver disease group samples with m6A

phenotype score Low group and High group in GSE135251 dataset were shown. The closer the AUC in the ROC curve is to 1, the better the diagnostic effect is. When AUC was between 0.5 and 0.7, the accuracy was low. When AUC was 0.7-0.9, it had a certain accuracy. AUC > 0.9 had high accuracy.

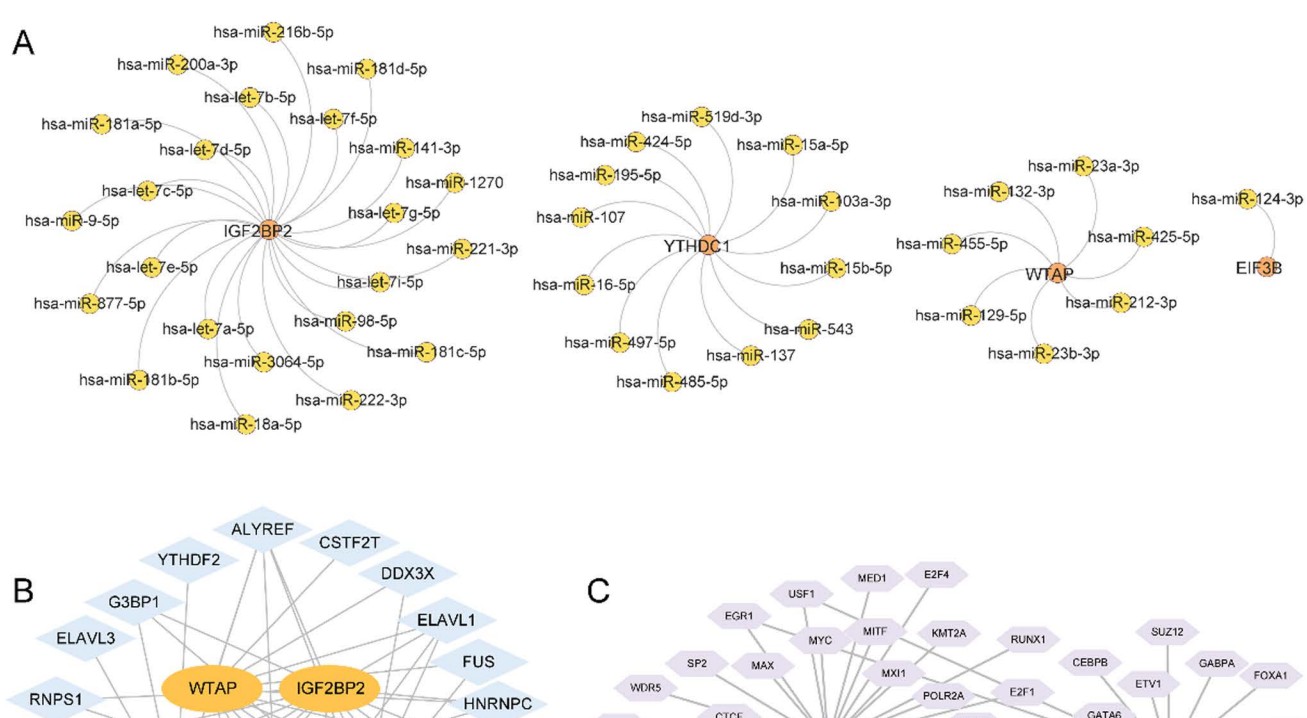

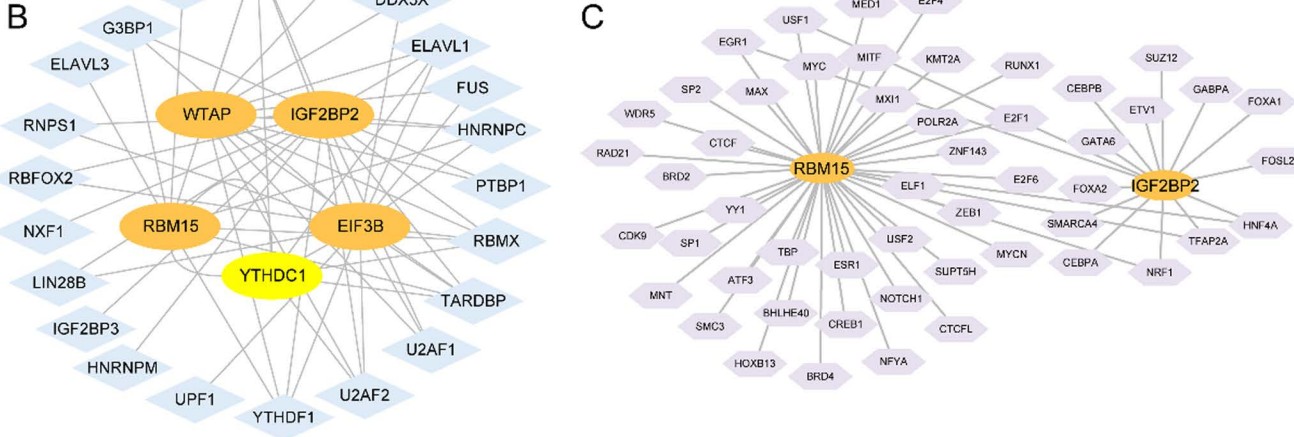

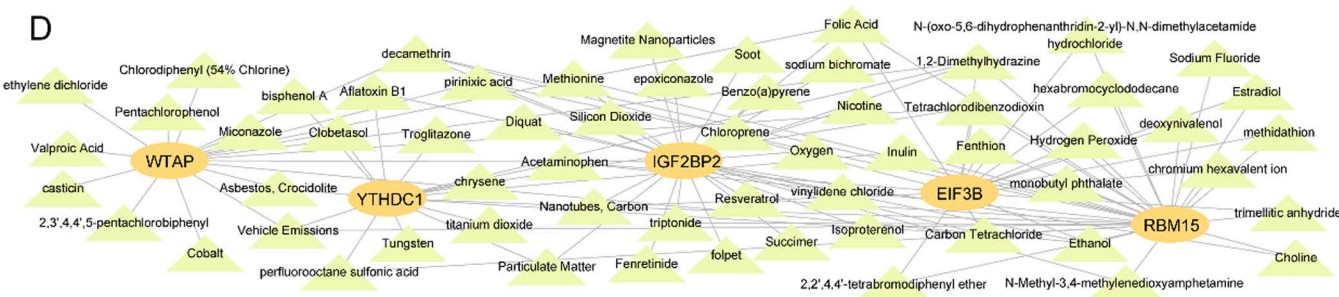

**Fig 8. Construct mRNA-miRNA, mRNA-RBP, mRNA-TF and mRNA-drugs Interaction Networks.** A. The orange dots in the mRNA-miRNA interaction network of key genes are mrnas; Yellow dots are mirnas. B. The orange oval blocks in the mRNA-RBP interaction network of key genes are mrnas; Blue diamonds are RBP; The yellow oval blocks are both mRNA and RBP. C. The orange oval block in the mRNA-TF interaction network of key genes is mRNA; Purple hexagonal blocks are transcription factors (TFS). D. The orange oval blocks in the mRNA-drugs interaction network of key genes are mrnas; Green triangles are specific molecular compounds (drugs). RBP: RNA binding protein; TF: Transcription factors.

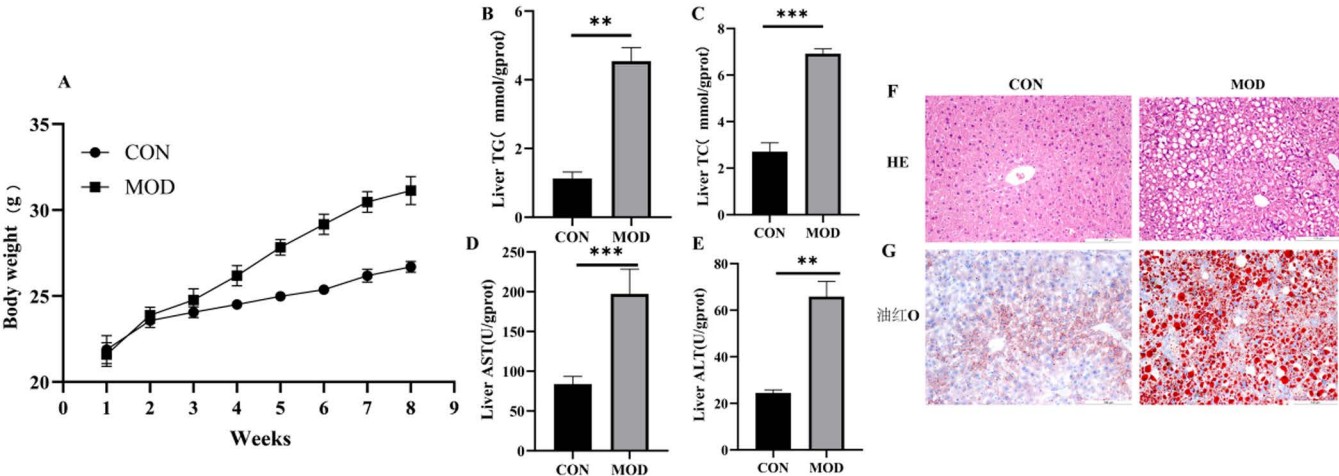

**Fig 9. High-Fat Diet-Induced Mouse Model of Non-Alcoholic Fatty Liver Disease (NAFLD)** (A) Body weight gain. (B) Liver TG content. (C) Liver TC content, (D) Liver AST content. (E) Liver ALT content. n=6 in each group. (F) H&E-stained liver tissues(X200). (G) Representative Oil Red O staining of mouse livers in each group (X200). *p < 0.05, **p < 0.01 and ***p < 0.001 compared with the control group (CON).

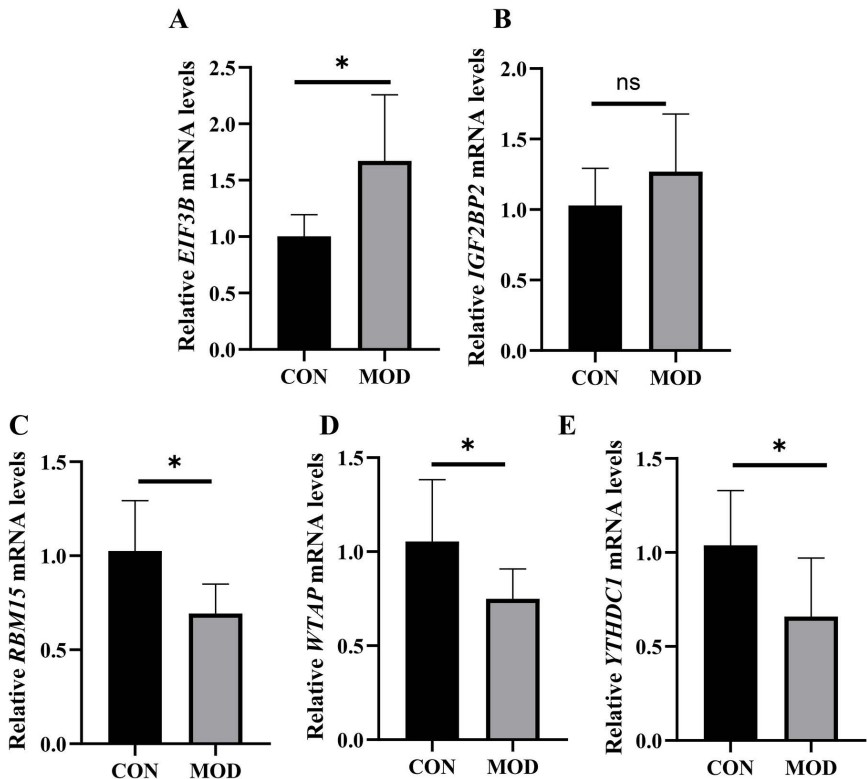

**Fig 10. Verification of The 5 Dignostic Hubgenes Using Real-Time Quantitative PCR (NAFLD samples = 6, control samples = 6).** *p < 0.05, **p < 0.01 and ***p < 0.001.

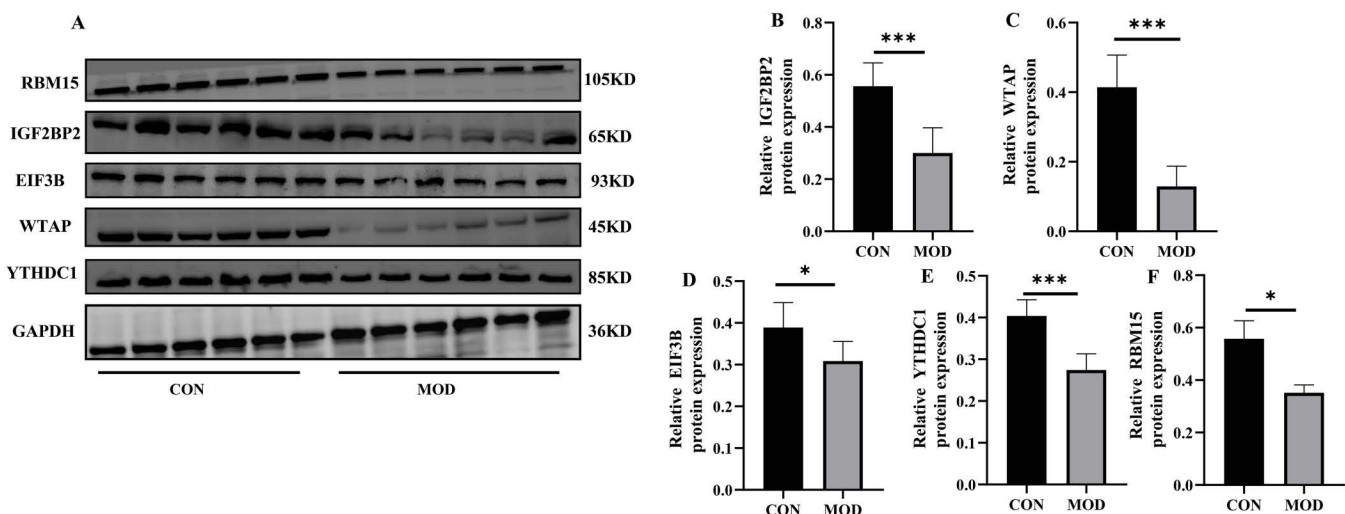

**Fig 11. Using Western Blotting to Validate 5 Significantly Changed Genes (NAFLD Samples = 6, Control Samples = 6).** *p < 0.05, **p < 0.01 and ***p < 0.001.

## Western blotting analysis

Compared to those in the control group, the levels of RBM15, EIF3B, YTHDC1, IGF2BP2 and WTAP proteins were significantly lower in the NAFLD group (Fig 11, full blot shown in S9 Fig).

## Discussion

Nonalcoholic fatty liver disease (NAFLD) is a common hepatic disease that substantially affects the quality of life of patients and is often associated with various complications. Although the specific pathophysiological mechanisms underlying NAFLD have not been fully elucidated, many studies have shown that genetic factors, immune dysregulation, and RNA methylation play key roles in its development. Specifically, m6A methylation, a form of RNA methylation, significantly affects the levels of genes related to lipid metabolism, insulin resistance, and inflammatory responses; thus, it is highly important for preventing and treating NAFLD. In this study, bioinformatics approaches were used to identify nine DEGs related to NAFLD from 34 m6A methylation-related genes, highlighting that m6A RNA methylation is important for NAFLD. The results of the GO analysis showed that the above genes were related primarily to the regulation of selective mRNA splicing, including RNA methylation. At the molecular function level, these genes were implicated in key activities, such as translation initiation factor activity, nucleic acid binding, and RNA binding. Based on these results, we established a diagnostic model using nine hub genes associated with m6A methylation (EIF3B, EIF3A, IGF2BP2, WTAP, YTHDC1, METTL5, ZNF217, RBM15, and YTHDF1). Among the biomarkers, YTHDC1 (AUC = 0.923) and WTAP (AUC = 0.912) showed superior diagnostic performance. This indicated good model performance and close alignment with the ideal predictive model, thus proving its robust diagnostic accuracy and providing a promising tool for clinical applications in NAFLD.

EIF3B, an important member of the family of eukaryotic translation initiation factors (EIFs), influences eukaryotic cell translation by modulating the interaction between ribosomes and RNA, serving as the rate-limiting step during protein biosynthesis. It can accelerate esophageal squamous cell carcinoma development by upregulating the expression of

TEX9 [17]. Additionally, in chronic myeloid leukemia, high EIF3B expression promotes cell proliferation and viability while suppressing apoptosis, thus facilitating the progression of the disease [18].

The m6A "reader" YTHDC1 found in the nucleus [19] has diverse functions. It can interact with nuclear export adaptor proteins (e.g., SRSF3) and splicing factors, facilitating the binding of RNA to SRSF3 and NXF1 and thus regulating m6A-modified mRNA nuclear export [20]. YTHDC1/ABCB6 is involved in cognitive dysfunction in AD mice by modulating neuronal ferroptosis. Moreover, in cooperation with PKM2, YTHDC1 affects the stability of the FOXO1 mRNA, subsequently influencing autophagy and disease alleviation. In renal cell carcinoma, YTHDC1 decreases mRNA stability of the downstream target gene ANXA1 in an m6A-dependent manner and decreases sunitinib sensitivity in renal cancer cells [13]. Conversely, in breast cancer, it enhances the stability of the EGF mRNA and increases doxorubicin resistance in breast cancer cells, thus promoting chemoresistance [21].

The m6A methyltransferase RNA binding motif protein 15 (RBM15) contains specific RNA recognition domains that enable it to directly bind to the RRACH-rich sequences of target mRNAs. By recruiting the WTAP-METTL3 complex, RBM15 modulates m6A RNA methylation and controls the activation of multiple pathways, including the Notch, Wnt, and JAK/STAT pathways [22]. RBM15 promotes the growth and migration of liver cancer cells and increases cancer cell growth, colony formation, invasion, and epithelial-mesenchymal transition [23,24]. The m6A modification of TMBIM6 mediated by RBM15 increases its stability in an IGF2BP3-dependent manner, promoting the development and progression of laryngeal squamous cell carcinoma [25]. RBM15 can also regulate YES1 m6A modification in an IGF2BP1-dependent manner, promoting the development of hepatocellular carcinoma [23].

Moreover, m6A modification can regulate the stability of inflammation-related transcripts, including proinflammatory and anti-inflammatory factors. In our study, specific m6A-associated genes were positively related to the degree of inflammatory cell infiltration, suggesting that m6A modification may modulate the inflammatory response in NAFLD by affecting the levels of inflammatory mediators. The complex interplay between immune cell infiltration and m6A modification is a central element of disease pathophysiology. Additionally, m6A methylation can regulate the stability of mRNAs of specific TFs and signaling molecules, thus affecting the differentiation of T-cell subpopulations such as Th1, Th2, Th17, and Treg cells [26]. These findings suggest that specific immune cell populations may exert a stronger influence on the disease process than others. The significant differences in the infiltration levels of immune cells, such as CD8+ T cells, suggest that these cell types may affect the development and progression of NAFLD.

Epigenetics has recently shifted the attention of researchers to microRNAs (miRNAs), which are small endogenous RNAs of about 21–23 nucleotides that specifically combine with 3'-UTRs in target gene mRNAs [14]. They play a key role in gene regulation by promoting mRNA degradation or inhibiting translation. They are cell-specific and organ-specific and are associated with various biological processes, such as cell differentiation, proliferation, embryonic development, and apoptosis. MiRNAs are closely associated with liver disease and regulate lipid metabolism. Lipid metabolism disorders are the core of NAFLD pathogenesis. The overexpression of miR-23b-3p inhibits the TGF-β1-induced proliferation and apoptosis of ASMCs, as TGFR2 is a direct and functional target [15]. Lowering miR-15b levels exacerbates cardiomyocyte apoptosis by downregulating Bc1–2 and MAPK3 [16]. PA stimulates macrophage production of miR-3064-5p to mediate metabolic inflammation in hepatocytes and adipocytes [27]; TNF-α induces miR-222 to regulate downstream protein expression or act as a new biomarker of liver fibrosis [28]. Overall,

miRNAs are important for gene regulation and disease pathogenesis and may serve as new markers of NAFLD. In the mRNA-drug network, fenretinide interacts with IGF2BP2, induces apoptosis, inhibits growth, and binds RAR with antitumor activity [29]. Resveratrol can regulate transporters and drug-metabolizing enzymes [30]. Pirinixic acid activates PPAR, lowers visceral fat and hepatic triglycerides, increases insulin sensitivity, and almost completely inhibits IL-1-induced IL-6 and prostaglandin production and COX-2 levels by suppressing the NF-κB pathway, which can interact with three key genes [31]. These findings suggest that these drug targets are related to the treatment of NAFLD and lay the scientific foundation for drug discovery.

To further investigate how m6A methylation affects NAFLD, we constructed a NAFLD mouse model by administering an HFD and experimentally validated five previously identified genes. Among these five key genes, the expression patterns of RBM15, YTHDC1, WTAP, and IGF3 BP2 were consistent, whereas the expression of EIF3B did not exhibit a clear trend. Further validation of these four significantly altered genes through Western blotting analysis revealed that three key genes (RBM15, YTHDC1, and WTAP) presented consistent changes at the transcriptional and protein expression levels, whereas the IGF2BP2 protein level did not change considerably. This difference occurred probably due to the complex interactions involved in the translation of cDNA into proteins, including transcription, translation, and posttranslational modifications. This was a retrospective, and based on the results obtained, we aim to conduct further in-depth research to assess the significance of the m6A phenotypic score for predicting the prognosis of NAFLD patients and its potential as a biomarker for disease progression. We also investigated the interaction of m6A RNA methylation with other epigenetic modifications in NAFLD to reveal new regulatory mechanisms.

While this study enhances the analytical rigor through multi-dataset integration and rigorous batch effect correction, several limitations in sample characteristics and methodological scope require acknowledgment. Although the aggregated sample size (N=294) provides initial statistical power, latent heterogeneity in clinical features (e.g., NAFLD progression stages, comorbidities) across cohorts may constrain model generalizability. The 5-gene signature (RBM15, IGF2 BP2, EIF3B, YTHDC1, WTAP), derived from SVM-LASSO combinatorial screening, demonstrated robust diagnostic performance in internal validation; however, its clinical translatability remains to be confirmed through independent external cohorts with expanded demographic diversity. Additionally, unmeasured confounders—including lifestyle factors (dietary patterns, physical activity) and concomitant medications—may introduce bias into m6A-related gene expression profiles, warranting covariate adjustment in subsequent studies.

Methodologically, while we successfully identified NAFLD-associated m6A regulators, the mechanistic underpinnings of their methylation regulatory roles remain partially unresolved. Notably, molecular dynamics (MD) simulations—a gold-standard approach for probing atomic-level biomolecular interactions—were not employed to delineate the structural dynamics of m6A protein-RNA binding interfaces. To address these gaps, future work will adopt a dual-strategy framework, first, Multi-center collaborations to establish validation cohorts with standardized clinical metadata, complemented by single-cell or spatial transcriptomics to dissect microenvironment-specific m6A regulatory circuits; Second, computational-experimental integration leveraging MD simulations to reconstruct free energy landscapes of m6A reader protein-RNA complexes, thereby bridging molecular mechanisms to NAFLD pathogenesis. This synergistic approach is anticipated to resolve current limitations in both biological interpretability and clinical applicability.

## Supporting information

**S1 Table. NAFLD Data Set Information List.**
(PDF)

**S2 Table. M6A-Related Genes List.**
(PDF)

**S3 Table. The Primer Information.**
(PDF)

**S4 Table. GO Enrichment Analysis Results.**
(PDF)

**S5 Table. mRNA-miRNA Interaction Network Nodes.**
(PDF)

**S6 Table. mRNA-RBP Interaction Network Nodes.**
(PDF)

**S7 Table. mRNA-TF Interaction Network Nodes.**
(PDF)

**S8 Table. mRNA-Drug Interaction Network Nodes.**
(PDF)

**S9 Fig. Uncropped Western blot images for blots used in** Fig 11.
(PDF)

## Acknowledgements

This work has greatly benefited from the contributions of GEO. We express our gratitude to the GEO network for generously sharing substantial amounts of data, and thank the Yunnan Provincial Engineering Research Center of PreventativeTreatment of Traditional Chinese Medicine, In addition, the authors would like to thank the editors and the anonymous reviewers for their valuable comments and suggestions to improve the quality of the paper.

## Author contributions

**Conceptualization:** Yan Qi, Jianguo Ma.

**Data curation:** Yangyang Fu, Jing Xu.

**Formal analysis:** Rongyi Xu, Renlin Li, Yangyang Fu.

**Funding acquisition:** Yan Qi.

**Methodology:** Renlin Li.

**Project administration:** Lei Zhou.

**Software:** Jing Xu.

**Supervision:** Yan Qi.

**Validation:** Rongyi Xu, Renlin Li, Jing Xu.

**Writing – original draft:** Jianguo Ma, Rongyi Xu.

**Writing – review & editing:** Yan Qi, Lei Zhou.

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
