## [Decision Letter · Decision Letter 0]

23 Feb 2025

PONE-D-25-03540Integration of bioinformatics and identification of the role of m6A genes in NAFLDPLOS ONE

Dear Dr. Qi,

Thank you for submitting your manuscript to PLOS ONE. After careful consideration, we feel that it has merit but does not fully meet PLOS ONE’s publication criteria as it currently stands. Therefore, we invite you to submit a revised version of the manuscript that addresses the points raised during the review process.

We look forward to receiving your revised manuscript.

Kind regards,

Jinhui Liu

Academic Editor

PLOS ONE

Journal Requirements:

3. Thank you for stating the following financial disclosure: Study design: the National Natural Science Foundation of China (Grant number [82460794]) , Xingdian Talent Support Program-Youth Talent Special Project (2023. No. 166).

Data collection and analysis:High-level TCM talents: (reserve talents) project of Yunnan (2021. No. 1)

Additional Editor Comments:

Authors should revise according to the suggestions of reviewers. The modifications should be marked. A point to point response letter is needed.

Reviewers' comments:

Reviewer's Responses to Questions

**Comments to the Author**

1. Is the manuscript technically sound, and do the data support the conclusions?

Reviewer #1: Yes

2. Has the statistical analysis been performed appropriately and rigorously? 

Reviewer #1: Yes

3. Have the authors made all data underlying the findings in their manuscript fully available?

Reviewer #1: Yes

4. Is the manuscript presented in an intelligible fashion and written in standard English?

Reviewer #1: Yes

5. Review Comments to the Author

Reviewer #1: They empolyed gene sequencing and animal model to demonstrate the diagnostic model based on the five key genes. The work is systematic and comprehensive.The study provides valuable insights into the molecular mechanisms underlying NAFLD and highlights the potential of m6A RNA methylation as a therapeutic target for this disease. I am not the expert for bioinformation, there are some suggestions from molecular interaction view to potentially improve the manuscript.

1. The sample size and potential data bias should be considered, for example in Fig 5ABC.

2. The detailed molecular mechanisms by which the identified genes regulate m6A methylation might be explored through MD simulations in further studies.

6. PLOS authors have the option to publish the peer review history of their article (what does this mean? ). If published, this will include your full peer review and any attached files.

**Do you want your identity to be public for this peer review?** For information about this choice, including consent withdrawal, please see our Privacy Policy .

Reviewer #1: **Yes: ** Jiebo Li

---

## [Author Response · Author response to Decision Letter 1]

7 Mar 2025

Comment 1: Thank you for reviewing my manuscript, I have carefully checked the article, checked the publication requirements of PLOS ONE journals, and confirmed that the format is correct.

2.To comply with PLOS ONE submissions requirements, in your Methods section, please provide additional information regarding the experiments involving animals and ensure you have included details on (1) methods of sacrifice, (2) methods of anesthesia and/or analgesia, and (3) efforts to alleviate suffering.

Comment 2: Thank you for reviewing my manuscript. To improve the transparency of the article, I have supplemented the full dose of the drug-pentobartal sodium used in line 217, page 10.

3.Thank you for stating the following financial disclosure: Study design: the National Natural Science Foundation of China (Grant number [82460794]) , Xingdian Talent Support Program-Youth Talent Special Project (2023. No. 166).

Data collection and analysis:High-level TCM talents: (reserve talents) project of Yunnan (2021. No. 1)

Comment 3: Thank you for reviewing my manuscript. The three projects mentioned in the article are all led by the corresponding author Yan Qi, and play the role of "project leader" in the three projects. The main role played by the funders was financial support and did not have additional roles in the study design, data collection and analysis, the decision to publish, or the preparation of the manuscript. I have written in my cover letter including this modification. Please change the online submission form on my behalf. We thank you again for your careful reading.

4.PLOS ONE now requires that authors provide the original uncropped and unadjusted images underlying all blot or gel results reported in a submission’s figures or Supporting Information files. This policy and the journal’s other requirements for blot/gel reporting and figure preparation are described in detail at https://journals.plos.org/plosone/s/figures#loc-blot-and-gel-reporting-requirements and https://journals.plos.org/plosone/s/figures#loc-preparing-figures-from-image-files. When you submit your revised manuscript, please ensure that your figures adhere fully to these guidelines and provide the original underlying images for all blot or gel data reported in your submission. See the following link for instructions on providing the original image data: https://journals.plos.org/plosone/s/figures#loc-original-images-for-blots-and-gels.   

Comment 4: Thank you for reviewing my manuscript. As for the requirements of image data, I have made the whole film of gel imaging according to the guidelines of the journal. In addition, I will upload the whole original uncropped and unadjusted images as an attachment for the reviewers and editors to view. Please confirm that these measures meet the requirements of the journal. Thank you again for your patient review.

5.Please review your reference list to ensure that it is complete and correct. If you have cited papers that have been retracted, please include the rationale for doing so in the manuscript text, or remove these references and replace them with relevant current references. Any changes to the reference list should be mentioned in the rebuttal letter that accompanies your revised manuscript. If you need to cite a retracted article, indicate the article’s retracted status in the References list and also include a citation and full reference for the retraction notice.

Comment 5: Thank you for reviewing my manuscript. We have verified all references against original sources to ensure formatting consistency (e.g., DOI, volume/issue, page numbers) and confirmed that all cited works are appropriately acknowledged., and that there are no retracted papers. Thanks again for your review.

Additional Editor Comments:

Authors should revise according to the suggestions of reviewers. The modifications should be marked. A point to point response letter is needed.

Reviewers' comments:

Reviewer's Responses to Questions

Comments to the Author

1. Is the manuscript technically sound, and do the data support the conclusions?

Reviewer #1: Yes

2. Has the statistical analysis been performed appropriately and rigorously?

Reviewer #1: Yes

3. Have the authors made all data underlying the findings in their manuscript fully available?

Reviewer #1: Yes

4. Is the manuscript presented in an intelligible fashion and written in standard English?

Reviewer #1: Yes

5. Review Comments to the Author

Reviewer #1: They empolyed gene sequencing and animal model to demonstrate the diagnostic model based on the five key genes. The work is systematic and comprehensive.The study provides valuable insights into the molecular mechanisms underlying NAFLD and highlights the potential of m6A RNA methylation as a therapeutic target for this disease. I am not the expert for bioinformation, there are some suggestions from molecular interaction view to potentially improve the manuscript.

1. The sample size and potential data bias should be considered, for example in Fig 5ABC.

Comment 1: We appreciate the reviewer’s concerns regarding sample size and potential data bias. In this study, our analyses utilized a relatively robust sample cohort: the NAFLD dataset comprised 47 NAFLD cases and 31 controls, while the GSE135251 dataset included 206 NAFLD cases and 10 controls, collectively providing sufficient statistical power. To mitigate batch effects and technical variability across datasets, we implemented rigorous batch correction using the sva R package, ensuring data homogeneity prior to integrative analysis. Principal component analysis (PCA) results (Figs. 2C-D) demonstrated effective harmonization of expression patterns post-correction, confirming successful mitigation of inter-dataset batch effects.

Furthermore, cross-dataset validation was inherently incorporated through our multi-cohort integration strategy, substantially reducing bias inherent to single-dataset analyses. During feature selection and model development, we employed a dual-machine learning approach: initial identification of six candidate genes through support vector machine (SVM) optimization for maximal NAFLD diagnostic accuracy (Figs. 5A-B), followed by refinement via LASSO regression to establish a parsimonious five-gene signature (RBM15, IGF2BP2, EIF3B, YTHDC1, WTAP) (Figs. 5C-D). This sequential methodology enhances the biological plausibility and analytical robustness of our findings.

We will explicitly address these methodological considerations and potential limitations in the revised Discussion section (Page 32-33, Lines 587-610) to strengthen the contextual interpretation of our results.

We sincerely appreciate the reviewer’s insightful comments.

2. The detailed molecular mechanisms by which the identified genes regulate m6A methylation might be explored through MD simulations in further studies.

Comment 2: We sincerely appreciate the reviewer's constructive feedback on our work.This study employed a bioinformatics-driven approach to systematically identify and characterize NAFLD-associated m6A-related differentially expressed genes (m6A-DEGs). Through multi-tiered analytical strategies - including Gene Ontology (GO) enrichment profiling, machine learning framework development, and experimental validation - we elucidated the potential regulatory roles of these genes in NAFLD pathogenesis. Our findings provide novel insights into the molecular underpinnings of NAFLD, with core results rigorously validated through RT-qPCR and Western blot analyses.

Regarding molecular dynamics (MD) simulations, we fully acknowledge their critical value in deciphering protein-RNA interaction dynamics. However, the current methodological focus was strategically prioritized toward high-confidence identification and preliminary validation of m6A regulatory targets, coupled with technical constraints in computational resource allocation. While this phase did not incorporate MD-based mechanistic exploration, we are committed to integrating this powerful tool in subsequent investigations to resolve atomic-level details of m6A reader protein-RNA binding thermodynamics.

We have incorporated this methodological perspective into the revised manuscript (Page 32-33, Lines 587-610)

Thank you for your thorough evaluation and valuable suggestions to strengthen the mechanistic depth of our research.

6. PLOS authors have the option to publish the peer review history of their article (what does this mean?). If published, this will include your full peer review and any attached files.

Do you want your identity to be public for this peer review? For information about this choice, including consent withdrawal, please see our Privacy Policy.

Reviewer #1: Yes: Jiebo Li

---

## [Decision Letter · Decision Letter 1]

11 Mar 2025

Integration of bioinformatics and identification of the role of m6A genes in NAFLD

PONE-D-25-03540R1

Dear Dr. Qi,

We’re pleased to inform you that your manuscript has been judged scientifically suitable for publication and will be formally accepted for publication once it meets all outstanding technical requirements.

Kind regards,

Jinhui Liu

Academic Editor

PLOS ONE

Additional Editor Comments (optional):

The authors have addressed the reviewers' concerns properly and revised the manuscript accordingly. The manuscript can be accepted for publication in its current form

Reviewers' comments:

Reviewer's Responses to Questions

**Comments to the Author**

1. If the authors have adequately addressed your comments raised in a previous round of review and you feel that this manuscript is now acceptable for publication, you may indicate that here to bypass the “Comments to the Author” section, enter your conflict of interest statement in the “Confidential to Editor” section, and submit your "Accept" recommendation.

Reviewer #1: All comments have been addressed

2. Is the manuscript technically sound, and do the data support the conclusions?

Reviewer #1: Yes

3. Has the statistical analysis been performed appropriately and rigorously? 

Reviewer #1: Yes

4. Have the authors made all data underlying the findings in their manuscript fully available?

Reviewer #1: Yes

5. Is the manuscript presented in an intelligible fashion and written in standard English?

Reviewer #1: Yes

6. Review Comments to the Author

Reviewer #1: The authors had replied the data bias error quesiton and molecular mechanism quesiton. I have no more comments.

7. PLOS authors have the option to publish the peer review history of their article (what does this mean? ). If published, this will include your full peer review and any attached files.

**Do you want your identity to be public for this peer review?** For information about this choice, including consent withdrawal, please see our Privacy Policy .

Reviewer #1: No

---

## [Editor Report · Acceptance letter]

PONE-D-25-03540R1

PLOS ONE

Dear Dr. Qi,

I'm pleased to inform you that your manuscript has been deemed suitable for publication in PLOS ONE. Congratulations! Your manuscript is now being handed over to our production team.

Kind regards,

on behalf of

Dr. Jinhui Liu

Academic Editor

PLOS ONE